# Infection Control in the Era of COVID-19: A Narrative Review

**DOI:** 10.3390/antibiotics10101244

**Published:** 2021-10-14

**Authors:** Nour Shbaklo, Tommaso Lupia, Francesco G. De Rosa, Silvia Corcione

**Affiliations:** 1Department of Medical Sciences, Infectious Diseases, University of Turin, 10126 Turin, Italy; francescogiuseppe.derosa@unito.it (F.G.D.R.); silvia.corcione@unito.it (S.C.); 2Department of Infectious Diseases, Cardinal Massaia, 14100 Asti, Italy; tommaso.lupia89@gmail.com; 3School of Medicine, Tufts University, Boston, MA 02111, USA

**Keywords:** infection control, COVID-19, healthcare-associated infection

## Abstract

COVID-19 quickly became a pandemic causing millions of infections and mortalities. It required real-time adjustments to healthcare systems and infection prevention and control (IPC) measures to limit the spread and protect healthcare providers and hospitalized patients. IPC guidelines were adopted and developed based on experience gained during the MERS-CoV and SARS-CoV outbreaks. The aim of this narrative review is to summarize current evidence on IPC in healthcare settings and patients with COVID-19 to prevent nosocomial infections during the actual pandemic. A search was run on PubMed using the terms (‘COVID-19’ [Mesh]) AND (‘Infection Control’ [Mesh]) between 2019 and 2021. We identified 86 studies that were in accordance with our aim and summarized them under certain themes as they related to COVID-19 infection control measures. All the guidelines recommend early diagnosis and rapid isolation of COVID-19 patients. The necessary precautions should be taken comprising the whole process, starting with an infectious disease plan, administrative and engineering controls, triage, and PPE training. Guidelines should target modes of transmission, droplet, aerosol, and oral–fecal, while recommending control precautions. Healthcare facilities must promptly implement a multidisciplinary defense system to combat the outbreak.

## 1. Introduction

A pneumonia outbreak of unknown origin, which was detected in Wuhan, Hubei province, China by the end of 2019, quickly became a global concern [1,2]. The responsible agent was found to be a virus from the *Coronaviridae* family [3]. The new virus was initially labelled 2019-nCOV and subsequently renamed SARS-CoV-2 due to its resemblance to the virus from the previous SARS-CoV pandemic. SARS-CoV-2 causes coronavirus disease 2019 (COVID-19) [4]. The virus started spreading globally, resulting in 171.6 million cases and 3.6 million attributed fatalities as of 3 June 2021 [4,5,6,7].

Infectivity is thought to begin before symptoms and decrease significantly around seven days after the onset of symptoms [8]. The infectious period is reported to depend upon the seriousness and stage of the patient’s infection [9].

COVID-19 poses significant social and health risks because even asymptomatic individuals can transmit the disease. Infected individuals who spread the disease through droplets during close contact situations are the main source of infection [8]. In healthcare settings, contact of the mucosae with infectious respiratory droplets or fomites was the most common pathway to human transmission [9]. In previous studies, however, sputum, nasal or nasopharyngeal secretions, endotracheal aspirate, broncho-alveolar lavage, urine, faecal matter, tears, conjunctival secretions, blood, and lung tissues were also found to transmit the virus [9,10,11,12,13].

Healthcare workers (HCWs) account for a significant rate of infections in these outbreaks because of multiple prolonged exposure [14]. In addition, stress, heavy workload, and sudden changes in the routine during the COVID-19 outbreak has made it difficult for many HCWs to apply preventive measures [15].

The high rate of infection among HCWs deserves further investigation. In a survey done among HCWs, the increased exposure to COVID-19 patients did not promote self-reported infection prevention and control (IPC) behaviours. HCWs who contacted confirmed and suspected COVID-19 patients even reported worse in some IPC. It may be due to a shortage in supplies, human deficiency, and high workload. This explains that the subsequent lack of protective equipment, human resources, and prompt assistance is critical in combating COVID-19 [16].

Lack of awareness, discomfort in PPE, poor knowledge of IPC or its negligence among staff influence pathogen spread. In addition, the presence of asymptomatic patients and reliance on IPC measures that do not account for all the dynamics of the pathogens may play a role in nosocomial transmission [17,18,19,20].

A policy rule is not the only factor in the IPC behaviours of HCWs. The outbreak, contact with confirmed and suspected patients, key clinical departments (such as intensive care unit and emergency department) are vital risk factors in the pandemic and affect nosocomial infections worldwide [19,21]. Other influencing factors associated with HCWs’ IPC behaviours include years of experience and preparedness [17].

New and unique infectious disease outbreaks are a danger to healthcare providers and other leading providers because of limited awareness of the emerging threats and reliance on IPC measures that do not account for all the dynamics of the new pathogens [13,14]. Theoretically, IPC prevents or stops the spread of infections in healthcare settings, making it pivotal to reducing the spread of the pandemic. In the case of COVID-19, IPC guidelines were adopted and developed based on experience gained during the MERS-CoV and SARS-CoV outbreaks, despite both viruses being from the same family of SARS-CoV-2; however, they share very low similarities [14,15,16].

The WHO has defined eight pillars for an IPC structure, namely: IPC programmes, IPC guidelines, education and training, and surveillance [22]. Others include multimodal strategies, monitoring and feedback, workload, staffing, bed capacity, built environment, materials, and equipment for IPC at healthcare facilities [17].

Poor point of care risk assessment is a factor contributing to the spread of COVID-19 among HCWs. Ensuring triage, early recognition, and isolation of patients with suspected COVID-19 is the first IPC strategy to prevent or limit transmission in healthcare settings [23].

The following critical healthcare IPC measures are required to prevent or limit COVID-19 spread in health facilities, including having the following in place: an IPC programme or at least a dedicated and trained IPC focal point, engineering and environmental controls, administrative controls, standard and contact precautions, screening, robust surveillance, and vaccination of health workers [18].

The aim of this narrative review is to summarize current evidence on IPC in healthcare settings and patients with COVID-19 to increase awareness among healthcare workers about infection control to prevent nosocomial infections during actual pandemics.

## 2. Methods

The current narrative review followed five steps: identifying the research question, search methods for identifying relevant studies, study selection, charting and summarizing data, and reporting the results.

The main research question was to summarize current evidence on IPC in healthcare settings and patients with COVID-19.

A search was run on PubMed using the terms (‘COVID-19’ [Mesh]) AND (‘Infection Control’[Mesh]) in English. Results were limited to 1 December 2019 and 1 October 2021. Studies were filtered for practice guidelines, guidelines, meta-analysis, systematic review, and review (Figure 1, Table 1).

A list of 692 papers was generated from the initial search. Then reviewers studied titles and abstracts. Secondly, 86 papers were included. Finally, quality assessment of full-text studies was performed by two independent reviewers (NS and TL). Researchers reviewed the summary of all articles sought, and ultimately used data from 86 full articles to compile this review paper. Researchers assessed for inclusion all titles and abstracts without language limitations in English. We included papers that described evidence on IPC in healthcare settings and patients with COVID-19. We excluded papers with no methods described, duplicated other studies previously included, not strictly related to COVID-19 pandemic and according to journal importance and number of references. Results were reported in twelve categories.

## 3. Results

### 3.1. Develop an Infectious Disease Plan

This includes assigning an infection control practitioner, training at-risk workers, avoiding adverse work events that promote the spread of infection, such as gatherings and long working hours, and applying teleworking where possible. Pandemic planning by hospital management and governments must be in parallel [24].

Adjusting working shifts may reduce COVID-19 exposure and balance personal protective equipment (PPE) consumption. Fewer personnel working slightly more hours may reduce the risk of exposure to multiple healthcare workers. If feasible, personnel should avoid working in both COVID and non-COVID units, as this may increase the risk of cross-transmission [25]. Figure 2 and Table 2 summarizes the IPC recommendations.

### 3.2. Administrative Controls

Administrative controls are policy, procedure, and process changes that decrease the exposure of individuals to a certain threat. They focus on ensuring that the proper prevention measures are taken, all work procedures are documented, administrative personnel are well trained to proper procedures, and that their use is enforced [26].

For suspected COVID-19 patients, it is essential from an infection control perspective to start contact and droplet precautions. All guidelines recommend cohorting suspected COVID-19 patients in a well-ventilated room when single rooms are not available [27]. During the COVID-19 pandemic, hospitals have placed administrative controls to almost every aspect of their operations. Hospitals that cohort confirmed COVID-19 patients also have implemented information on safe administrative controls to increase the efficacy of that policy [25].

Elective surgeries and noncritical visits should be postponed. For respiratory symptoms, triage protocols on admission must be followed. Triage should restrict entrance to the hospital with temperature monitoring. Hospital entry points, waiting rooms, and patient rooms should have hand disinfectants with 60–95% alcohol and no contact waste containers. A physical barrier of plastic or glass must be placed to isolate triage personnel [28].

### 3.3. Assessment Tool for Triage

Since the turnaround times of COVID-19 testing can be long, patients should be triaged to different wards according to their clinical history and likelihood of developing COVID-19. Patients with no COVID-19 symptoms should be separated from patients with low or high suspicion of COVID. Confirmed COVID-19 cases should be cohorted in an isolated ward. In this way, positive and negative patients are separated as early as possible. Daily assessment should be made to confirm that incubating infections are identified and isolated [29].

### 3.4. Engineering Controls

a.Viral Clearance Periods

Room ventilation eliminates viral aerosols fairly quickly. Every air exchange removes around 63% of the virus [30]; after x room exchanges, the residual viral load is 0.37×. If there are 12 air exchanges in an hour, five exchanges reduce the viral load to <1% in half an hour in an ICU. In general wards, there should be around six air changes per hour [31].

All the guidelines recommend performing aerosol generating procedures (AGPs) in negative pressure rooms using contact or airborne precautions [31]. However, they are less likely to be available during the pandemic. Well-ventilated rooms, with a high rate of air exchange, are more important for COVID-19 control than positive or negative pressure [31]. The Chinese guide also recommends air disinfection using air or pressure steam sterilizers. Sterilizing or incinerating patients’ clothes and bed sheets are included in several guidelines. Although faecal–oral transmission of COVID-19 has not been confirmed, the Chinese guide recommends disinfecting septic tanks. Only the ECDC, Chinese, and UK guidelines stress decontaminating transportation vehicles and trollies that carry COVID-19 patients [32].

b.Theatre bundling

This is a common engineering control to reduce hospital-acquired infections and protect healthcare personnel [9]. Clean areas are defined as zones where no contact with any source of COVID-19 has occurred. These include staff and PPE donning rooms. Buffer (decontamination) areas are transition rooms between the clean and the potentially contaminated zones. This permits an additional barrier against cross-transmission within zones. PPE should be donned before entering the buffer zone, and it should be doffed before moving to clean zones. Potentially contaminated zones are all areas dealing with COVID-19 patient care. The theatre bundling concept using buffer areas has recently been accounted for because of its effectiveness [25].

c.Physical barriers

The World Health Organization (WHO), Center for Disease Control and Prevention (CDC), and Australian guidelines stress on engineering controls to manage patients such as: partitions in triage areas, curtains around each bed, sealed suctioning systems in intubated patients, and airflow management. The CDC recommends installing physical barriers of glass or plastic in the hospital reception area [32].

### 3.5. Aerosol Box

Sorbello et al. described a transparent plexiglass barrier aimed at decreasing the diffusion of aerosols during intubation. They showed several aerosol spread patterns and possible transmission to personnel performing AGP. Similar barriers have been adopted worldwide in healthcare facilities [32,33]. It has been suggested that when adequate PPE is not available, physical barriers might prevent exposure of personnel. However, the variability of methods, small sample size, scarce patient data, and low evidence on decreasing transmission pose many questions about the validity of the findings. In the absence of clear evidence, aerosol barriers are not recommended. They increase the working load; add obstacles to proper airway management; may be reservoirs for contact transmission; may damage PPE and, primarily, do not eliminate all aerosols [34].

### 3.6. Personal Protective Equipment (PPE)

Effective PPE programmes must include training, selection, and proper disposal of all PPE. The PPE recommended include fluid-resistant gowns, gloves, face shields, eye protection, hair cover, and N95 masques. Scrubs should be worn under the PPE. Shoes should be impermeable. Shoe covers can increase the risk of self-contamination during doffing [35]. Healthcare facilities should have protocols for proper donning and doffing of PPEs in a sequenced manner.

Studies have shown high rates of doffing mistakes, even with basic PPE, causing contamination. PPE doffing must start in the contaminated area with hand hygiene. Next, remove the face shield, shoe covers, gloves, gowns, goggles, cap, and surgical masque [36]. Donning and doffing PPE would take about 30 min each. The PPE that should be used for HCWs, according to the procedures reviewed, have been defined by the World Health Organization (WHO) [37].

### 3.7. PPE Reuse

Considering the worldwide shortage of PPE, the recent guidelines recommended surgical masques as an acceptable alternative to N95 masques, saving N95 for AGPs [32]. All guidelines discouraged reuse or extended use of single-use masques, except in critical scarcity when other strategies have been applied, such as reducing the need for PPE by engineering, administrative controls and supply management. The guidelines favoured extended use over reuse because of the reduced risk of contamination. All guidelines recommended checking for damage and testing the fit and seal before reuse [38].

For example, disinfecting PPE with UV-C is probably effective against COVID-19. This approach could provide enough energy to decontaminate the viral agent and maintain the equipment’s efficiency for reuse. However, more evidence is needed to assess the effectiveness of UV-C on PPE decontamination [39].

### 3.8. Types of PPE Masques

The terms filtering facepiece FFP1, FFP2, and FFP3 are used for high-performance filtering masques. Filtration is done by a web of polypropylene microfibers and electrostatic charges. Every protection factor indicates the extent to which the masque will eliminate hazardous agents. For FFP1, FFP2, and FFP3, the filtration is 80%, 94%, and 99%, respectively [38].

Fluid-resistant masques are a necessity. N95 is currently recommended for HCWs who have close contact with COVID-19 patients and perform AGP [39,40]. The masque can be used while visiting multiple patients for no more than 4–6 h, if it is not damaged or moistened [41].

### 3.9. Powered Air-Purifying Respirators (PAPRs)

To filter gases, PAPRs have a battery-run motor that sucks particles through a filter and cartridges. The filtered air is then delivered under positive pressure to a face piece. The positive pressure reduces leakage of possible contaminants. In contrast to N95 masques, PAPRs provide more protection. They have an assigned protection factor (APF) of 25 [39,42]. In addition, they eliminate the fit problem and can be worn with eyewear. Simultaneous use of an N95 for further protection is controversial [43]. N95 along with PAPR has been recommended during AGP to enhance protection and mitigate risk in cases of PAPR mechanical failure [43].

### 3.10. Environmental Cleaning

Cleaning environmental surfaces and equipment with disinfectants that are commonly used in the hospital is adequate and effective [41]. COVID-19 is sensitive to ethyl alcohol, povidone-iodine, sodium hypochlorite, benzalkonium chloride, hydrogen peroxide, and cresol soap [44]. All surfaces, including floors, walls, and objects in COVID-19 rooms, should be disinfected. A solution of 1000 mg/L chlorine is commonly used. Disinfection should be done three times a day and whenever contamination occurs. Cleaning personnel should wear appropriate PPE [45]. After the patient is discharged, windows should be left open for an hour prior to cleaning [28].

The use of laminar flow ventilation with UV-C and plasma treatment for room disinfection are considered feasible [46]. Walker et al. showed the efficacy of UV-C with a laminar flow device for viral inhibition. However, the study did not ensure effectiveness against types of coronavirus [47]. Wang et al. showed that environmental cleaning along with plasma air treatment are beneficial against COVID-19 [48]. The advantage of the synergy is that UV-C might not reach all surfaces because of shadows and overlapping objects. Similarly, cleaning is susceptible to human failure and missed spots [49].

### 3.11. Infectivity: Discharge and Quarantine

In a review of 17 studies of COVID-19 cultures, mildly to moderately ill patients were highly unlikely to be contagious after day 10. Immunocompromised and severely to critically ill patients may be contagious for more than 10 days. This evidence may be used to inform guidance on the duration of isolation [41].

a.Semi-quantitative Polymerase Chain Reaction (PCR)

For both qualitative and quantitative Real Time (RT)-PCR assays, the correlation between Ct values and the amount of virus in the original specimen is imperfect. It is therefore problematic to infer any relationship between an individual patient’s Ct value and their viral load. Ct values can also be affected by factors other than viral load. For example, if the specimen is not collected or stored properly or if the specimen is collected early during the infection, the Ct value may be higher than it would be under ideal conditions. Also, Ct values are not directly comparable between assays. Therefore, the use of Ct should be managed by an infectious disease specialist along with all the clinical and microbiological information as an adjunctive tool to manage suspected cases, knowing the low level of evidence in determining infectivity. Perhaps, the latter use of Ct values is considered in sub-acute, chronic, relapsing, and remitting courses to rule out infectivity [44].

b.Discharge and Quarantine

Patients should be advised to quarantine at home, minimize social interactions, and wear a masque when interacting. Patients may stop isolation seven days after the symptoms begin [26,50]. An admitted patient may be de-isolated after fever resolution without antipyretic medication, improvement in symptoms, and one negative PCR result or two in selected cases at least 24 h apart [50]. 

c.Antigen Testing for Nosocomial Infections

Antigen tests suffer from lower sensitivity compared to RT-PCR. However, they come with many advantages such as being robust, rapid, and easy-to-perform [51]. They showed a high sensitivity and specificity in respiratory samples of patients in the first week of COVID-19 [52]. Numerous studies on antigen testing compared to RT-PCR results using various tests were published [53,54]. The majority of the tests were within a pooled sensitivity of 70–90%. Therefore, they have the potential to become an important tool for the early diagnosis of COVID-19, especially in situations with limited molecular testing [52].

The WHO recommended the use of COVID-19 antigen tests for admitted patients when nucleic acid amplification test is unavailable or where prolonged turnaround times inhibit clinical utility, and within the first 5–7 days of symptoms [55]. The WHO sets the minimum performance requirements of 80% sensitivity and 97% specificity (ideally ≥99%) compared to the RT-PCR reference assay [55]; the ECDC agrees with these ranges but highlights that the aim should be at least 90% sensitivity [56]. Currently, the main testing worldwide depends on RT-PCR, although several countries such as, Germany, France, or United Kingdom, have already adopted antigen testing in their strategies [56].

d.Decision-making for pursuing work for HCW

In certain cases, PCR testing is recommended for returning to work in infected healthcare personnel if their PCR is negative before the isolation period is over [57]. German guidelines recommend healthcare workers who were hospitalized for COVID-19 to have two negative PCR tests at least 24 h apart before returning to work [58]. In Switzerland, re-testing infected healthcare workers after the isolation period is necessary for those working in high-risk units such as haemato-oncology, ICUs, and transplant units. There is some doubt about knowing the transmissibility from PCR results; therefore, the effect of repeated testing to decide the return of healthcare workers is uncertain [59].

### 3.12. Corpse Handling and Management

Standard precautions are recommended by almost all the guidelines when handling corpses. Only the UK, Australian, and Chinese guidelines recommend using body bags. The Chinese guidelines advise covering the nose, ears, mouth, anus, and open wounds of the dead body with gauze. A burial ritual can be permitted with standard precautions. A dedicated mean is recommended for corpse transport [30,44].

## 4. Discussion

To decrease exposures to COVID-19, all of the guidelines recommend early diagnosis and rapid isolation of COVID-19 patients [32]. The necessary precautions should be taken that comprise the whole process, which start with an infectious disease plan, administrative and engineering controls and triage [28]. Further, procedures such as elective surgeries and routine follow-ups should be postponed. Rooms should be equipped with hand disinfectants and proper ventilation [50]. Staff should be trained for appropriate donning and doffing of PPE [53].

### 4.1. Discrpancy and Limitations among International Guidelines

All international and national guidelines (CDC, WHO, ECDC, DHA, BDPCC, and PHE) agreed on the basic practices but with different specifications. For example, the Australian guidelines recommends training on IPC to ICU staff only. The CDC recommends providing surgical masks to patients depending on the area and risk assessment while surgical masks to staff only if N95 respirators are not available. The use of spatial separation between patients are recommended by the CDC, WHO, and DHA at 1-m distance, 2-m and according to state-level policies, respectively. Placing known or suspected patient in AIIR/negative pressure room is recommended by the ECDC while advised by the DHA, BDPCC, and PHE only when available. Cleaning and disinfection electronic equipment is recommended by ECDC and DHA only. The use of clean cloth towels is recommended by only the ECDC when paper towels are unavailable. Moreover, the global shortage of N95 and facemasks had forced the CDC and BDPCC to loosen their recommendations for face protection of healthcare personnel and recommend cloth masks when both are unavailable [53].

Discrepancy is also present regarding the disposition of patients after recovery and isolation precautions. The WHO guideline recommends continuing the precautions until a patient is asymptomatic. However, one study identified prolonged shedding of COVID-19 after recovery [53]. In addition, low- and middle-income countries often apply international IPC guidelines according to their local context. Therefore, the guidelines should consider the global context when recommending IPC measures [22].

A study has assessed the feasibility of COVID-19 infection control practices. The guidelines were not always feasible because of overcrowding in the emergency room leading to less commitment of the providers, insufficient training, lack of policy, and shortage of infection control materials [32].

### 4.2. Changes in Guidelines

Conflicting and changing recommendations have fueled the arms race among the facilities. Hospitals are spurring to adopt more strict infection control practices that often exceed the standards published by the CDC and WHO. Whenever a hospital adopts a new practice that is thought to be more protective, another feels pressure to follow. Examples include testing asymptomatic patients more frequently; using face shields and N95 respirators regardless of positivity and symptoms; additions to AGP list; and strict PPE, including shoe, hair, and leg covers [60].

In February, the WHO recommended contact and droplet precautions while reserving N95 respirators or PAPRs for AGPs. The CDC initially recommended N95 respirators but changed to medical masks in times of N95 shortages. This shift impresses that CDC guidance is driven by supply availability rather than science [60].

In fact, the CDC recently updated their guidelines to recommend that areas with high prevalence of COVID-19 should consider using N95 respirators in all asymptomatic patients undergoing AGP, regardless of test results [60].

The updated UK Health Security Agency (UKHSA) guidance consists of three categories focused on changes to the requirements for physical distancing, pre-elective procedure testing, and enhanced cleaning. In terms of physical distancing, UKHSA has recommended a reduction of physical distancing from 2 m to 1 m with appropriate mitigations where patient access can be controlled. UKHSA has also proposed removing the need for a negative PCR and 3 days self-isolation before selected elective procedures as currently advised by the National Institute for Health and Care Excellence (NICE). UKHSA has also encouraged re-adopting standard rather than enhanced cleaning procedures in low-risk areas [61].

### 4.3. Unanswered Questions

The evidence for using a type of filtering masque (FFP2, FFP3, N95) over a surgical masque is not robust, with a lack of clear evidence of the efficacy of high filtration masks. Variables such as fitting, testing, and personal use probably contribute to this uncertainty. The classification of AGP and risk of transmission from each procedure also are not clear yet. In addition, evaluation of the possibility of decontaminating and reusing N95 masks has been undertaken, with early results suggesting promise for both steam and UV sterilization. However, these results cannot be widely applied because of loss of filtering capacity [31,58]. The use of FFP2 masks by the public is also controversial. Some countries, such as Spain, has banned their use by the public to preserve them for healthcare workers [54].

The guidelines do not describe the use of protective hoods or headgear, though they are widely used in some countries [62]. Furthermore, there is slight evidence that double gloving in AGP may provide enhanced protection and reduce fomite contamination [63]. Facilities reported low rates of HCW infection after performing further precautions, such as careful doffing of PPE, showering, and oral and nasal disinfection [64]. In some cases, staff were isolated from families and kept under surveillance [65]. It is unknown if these extreme precautions are necessary or practical to prevent HCW infection.

A Cochrane review about PPE and protection of healthcare staff in contact with contaminated body fluids does not show a robust evidence with an almost complete absence of clinical studies examining relevant clinical outcomes. All interventions studied were supported by no more than one study and rated as low evidence [30,55]. For example, the review reports that gowns provide more protection than aprons; doffing supported by verbal instructions reduces errors; and air-purifying respirators may reduce contamination compared with conventional PPE [66].

The presence of virus in stool indicates the transmission through fecal–oral or fecal–droplet routes [31,67]. In recent studies, COVID-19 was detected in toilet bowls, air, and sinks [49]. One study detected the organism in air samples of hospital toilets, remaining viable for at least 30 min after flushing. This evidence suggests the possibility of fecal–droplet transmission [68]. Specific recommendations are needed regarding the prevention of fecal transmission in hospitals.

Studies have showed that rapid diagnosis is challenging because the manifestations of COVID-19 are nonspecific and may be confused with other microbial infections [69]. Routine testing is suggested by at least in one international practice statement [70], however it depends on the local resources of the hospital and the phase of the pandemic.

Regarding virulence factors, the recommendation of social distancing in the guidelines varies between 1 and 2 m; however, a recent study has shown that COVID-19 can travel more than 4 m [71]. Moreover, environmental influences, such as humidity, air flow, conditioners or fans, may affect the spread of droplets. An outbreak of COVID-19 related to conditioners was described in China [72]. These results suggest that the recommendations for spatial separation need to be reviewed again.

## 5. Conclusions

IPC measures should consider COVID-19 to spread as a droplet, aerosol, and oral–fecal route. Guidelines should target these modes of transmission while recommending control precautions. Awareness among healthcare workers about infection control is critical to prevent nosocomial infections and control outbreaks. During epidemics, healthcare facilities must promptly protect healthcare workers and patients and implement a multidisciplinary defense system to combat the outbreak. On the long term, more evidence-based infection control strategies are needed to uniform the guidelines. This could be achieved by understanding the transmission dynamics better, defining an evidence-based list of aerosol generating procedures, improved perception of the negative predictive value of both PCR and serological tests to decide discharge and organize staff to minimize risk and PPE use. The efforts should address the gaps that occur between the development of IPC guidelines, their introduction and implementation.

## Figures and Tables

**Figure 1 antibiotics-10-01244-f001:**
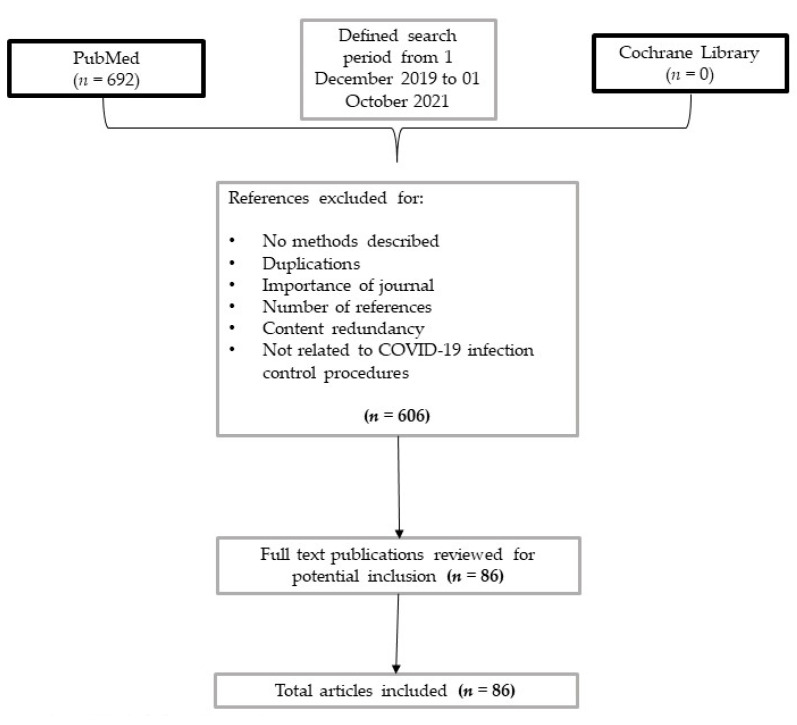
Flow-chart of studies screened and included in the analysis.

**Figure 2 antibiotics-10-01244-f002:**
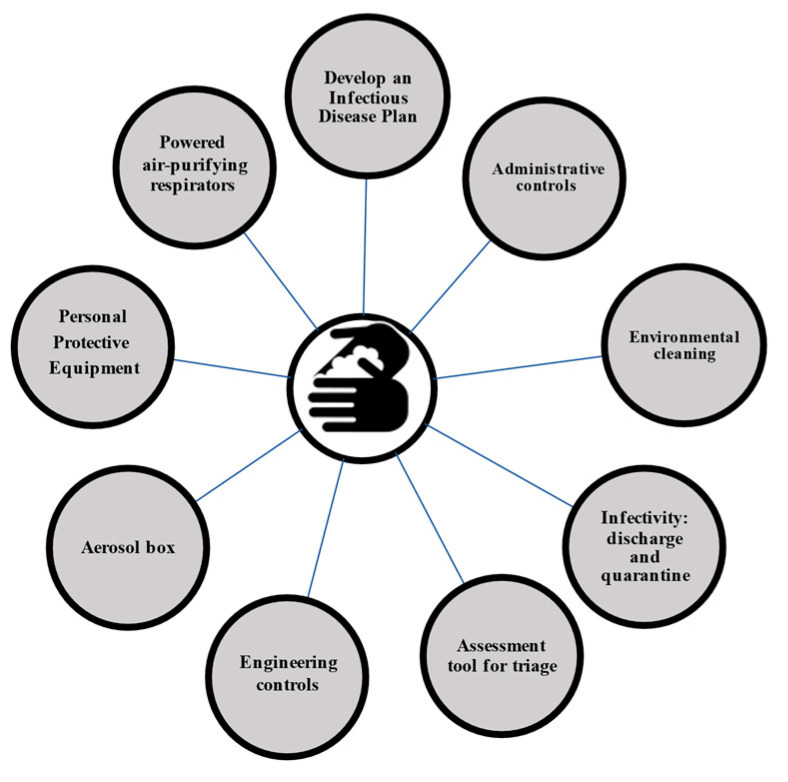
Infection prevention and control cornerstones in COVID-19.

**Table 1 antibiotics-10-01244-t001:** Types of included studies (guidelines and empirical research).

Topic	Number of Assessed Studies
IPC bundles	36
PPE	21
Disinfection and filtration techniques (UV, negative pressure, heat)	16
IPC for special procedures (tracheostomy, broncoscopy, endoscopy, CT scan)	8
ICU & LTCF	5

**Table 2 antibiotics-10-01244-t002:** Summary of IPC recommendations for COVID-19.

Recommendation	Description
Infectious Disease Plan [17,18]	Assigning a responsible infection control team. Education and training of health-care workers. Adjusting working shifts. Supplying PPE.
Administrative Controls [17,29]	Postponing elective surgeries and noncritical visits. Triage protocols on admission. Monitoring temperature on entrance.
Triage [21]	Triaged based on clinical history and infection susceptibility. Cohorting COVID-19 cases in an isolated ward. Daily assessment to identify incubating infections.
Engineering controls [18,22]	Maintaining viral clearance periods. Theatre bundling and transition areas. Physical barriers.
Personal Protective Equipment [30,31,36]	Training, selection, and proper disposal of all PPE. Sequenced PPE donning and doffing. Fluid-resistant gowns, gloves, face shields, eye protection, hair cover, and N95 masques.
Environmental cleaning [15]	Cleaning environmental surfaces and equipment with effective disinfectants.

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
