# Peer review of "Infection Control in the Era of COVID-19: A Narrative Review"

_antibiotics, 2021, doi:10.3390/antibiotics10101244_

Round 1
Reviewer 1 Report
The introduction should be improved, it is too brief and does not adequately address the interest of the study and its rationale. The first thing to do is to contextualise the object of the study.
Methods. It is essential for the publication of this article to have a methodology section where the method used to arrive at the results offered is explained. It is not explained whether the included studies were peer-reviewed and whether there were conflicts between reviewers.
Further development of the methodology is needed.
Author Response
Dear reviewer,
Thank you for this comment that improves notably our manuscript. According to your suggestion, we have added new paragraphs to the introduction section focusing more on: how health care workers are at high-risk for infection and their awareness and knowledge about infection control during the pandemic. Moreover, we added how health-care facilities should contain the outbreak in
a multi-disciplinary approach. You can find the changes highlighted in the Introducion section.
For the second comment, we agree with you. We have defined better the methodology and the steps of this current narrative review. We have improved “Methods” section of this manuscript according to your suggestions. You can find changes highlighted in the text.
We appreciate your extensive revision.
Best regards,
Dr. Nour Shbaklo

Reviewer 2 Report
Infection Control in the Era of COVID-19: A Narrative Review
This article reviews recommendations in relation to the transmission of infections.
The beginning of this article (in my opinion) refers to infection control in a healthcare facility;
Hygiene rules, eg hand hygiene, cleaning techniques, etc. are recognized recommendations;
I am interested if anything in these rules has changed since the pandemic was declared;
Have the institutions responsible for sanitary and epidemiological supervision introduced new guidelines? Has the epidemic forced a change / update of procedures?
Then the authors move on to the issue of testing, and I do not know if these recommendations apply to patients admitted to the ward or to the general population? Quarantine Policy - Does this apply to a mildly symptomatic patient who does not need to be hospitalized? Or an asymptomatic man who took the test for various reasons. After reading the entire manuscript, I know it deals with the control of nosocomial infections, but maybe it is worth highlighting this on lines 228-234 so that the reader is not confused? (suggests)
Preventive vaccinations play an important role in the prevention of infectious diseases. Is vaccination against COVID-19 recommended for healthcare workers as mandatory? What about vaccinating patients? - for example, should a relaxation of the rules on isolation / segregation of patients be considered when they have been vaccinated? Should vaccinated patients be tested? And how can any such recommendations relate to the problem of transmission of the virus by the vaccinated person (does vaccination protect only the vaccinated person or also protect against the spread of the virus?)
In my opinion, the manuscript is well written, and taking into account the above comments will complete the topic.
Author Response
Dear reviewer,
Thank you for this comment that notably improves our manuscript.
-We have improved introduction section to better focus the attention of the reader on manuscript aims. Moreover, we have improved the section “Changes in guidelines” updating references until 1 October 2021. You can find changes highlighted in the text.
-Regarding the testing, recommendations shall be applied to patients admitted. We have modified in the text accordingly to your suggestions.
- Regarding the quarantine, it can be applied both to mildly symptomatic and asymptomatic. Patients with mild symptoms are referred to territory health care facility and are kept under medical surveillance and following quarantine policies as well as asymptomatic.
- Regarding specifying nosocomial infections, although we are not sure which section you are referring to, we have specified this on the given lines.
- Vaccination against COVID-19 is mandatory for HCWs in many countries in Europe. In most of hospitals in Italy we currently test both RT-PCR and antigen tests for all patients admitted to emergency department regardless of vaccination. Moreover, weekly surveillance is mandatory in many hospitals for HCWs with RT-PCR and/or antigen tests. No relaxation was observed in my opinion. The WHO still recommends that vaccinated persons should continue to adhere to public health and social measures, including in health facilities and advises that all IPC measures for COVID-19 in health facilities be maintained for vaccinated health care workers, as well as unvaccinated workers. Recommendations according to transmission by the vaccinated person it is not possible to define yet.
We appreciate your extensive revision.

Reviewer 3 Report
This manuscript is timely and will be of interest to Antibiotic readers. The minor concerns with the paper include:
- each sub-section needs 1-2 sentences to describe the context (i.e. administrative controls) of the section with respect to the scientific community and/or COVID. As written these sections come across as jargon and may not be accessible to all readers.
- each sub-section could include a brief description of the number of papers that were discovered (Table 1) along with information as to the type of papers (primary data, reviews, recommendations) to help provide greater context.
- the narrative-style review as written doesn't reference the primary data collection and/or describe these data. Rather there is a summary of these results without describing the experimental design and limitations of these studies.
- table with recommendations that summarizes the paper results along with relevant citations. this would be helpful as a guide for others to help with infection control.
Author Response
1.
Dear Reviewer,
1. Thank you for this comment. We have added a description about administrative controls to introduce it to the reader in a simplified way before relating it to COVID. Please find the addition highlighted in text.
2. We have improved the methods section and methodology of this narrative review. Furthermore, we have Figure 1 and Table 1that summarize the papers included and excluded from the manuscript. Each section has statements referenced and a new table (Table 2) was added to summarize IPC recommendations for COVID-19 with the respective references.
3. We narratively review all the literature for IPC and COVID-19 and our aim was summarize pivotal elements of this topic. We have provided references for every statement, section, and subsection. We have described all the topics of IPC in a narrative way to make easy for the reader to read, raise more commentaries about the topic and address research questions in future. We hope that this reply could be worthy of your comment.
4. We have provided a new table (Table 2) with a summary of IPC recommendations with relevant citations according to your suggestions.
We appreciate greatly your extensive revision.
Best regards,
Dr. Nour Shbaklo

Round 2
Reviewer 1 Report
The indicated corrections have been made.
This manuscript is a resubmission of an earlier submission. The following is a list of the peer review reports and author responses from that submission.
Round 1
Reviewer 1 Report
In my opinion, the aim of this work is interesting, however the review did not adhere to the PRISMA guidelines. No details on study selection were reported (e.g., the PRISMA flow diagram is missing). The authors should provide at least a table with the general characteristics of each study. The risk of bias was not assessed. Please fully revise the manuscript following the PRISMA guidelines.
Reviewer 2 Report
This article is an overview / narrative scoping review on an important topic, Covid-19 infection prevention; unfortunately, I believe it is out-with the scope of the journal ‘Antibiotics’ and would recommend re-submission to a journal which focuses on the processes of infection prevention and control. If doing so, the authors may wish to provide further detail of the review methods applied, using recommended reporting methods, particularly in relation to quality appraisal of included studies so that the strength of evidence for recommendations is clear. In the results, an overview of the types of included studies (e.g. guidelines, empirical research etc), from which countries/contexts, and how many studies focused on which aspects of IPC would be interesting. It would also be helpful in the discussion to indicate what, if any, limitations have been found in the international guidelines included in the review i.e. WHO, ECDC, CDC, Cochrane reviews i.e. what does this scoping review add to what is already known.
Reviewer 3 Report
The authors presented in their review different measures which were described in the various papers and web pages published from 1 December 2019 and 3 April 2021.
Strength:
The paper is well structured
Actual topicGood stylistic and language
Limitation:
The situation is dynamic and these results might not be true in the current situation.
Different countries apply different measures. E.g. line 162, the use of FFP1, 2 and 3 masks are recommended. However, the citation is from France, in other countries are FFP3 masks not allowed according to my knowledge. This point is I think crucial - each country applies different rules, some rules are general for regions or continents. The authors summarized all measures without a note that some of the rules (e.g. use of FFP2 mask) are only applying in certain countries.
Method part is according my opinion not sufficient. PubMed is an excellent database, however, contain only a limited number of journals. Also to use only two terms I think it is insufficient. The citation of PubMed is missing. Also, the webpages such as RKI are cited in the text, they were for sure obtained from another source than PubMed.
Wrong citation style, the accession date is missing in every cited web-page, majority of the web-pages the authors cited are not anymore available so it is hard to verify if the citation is correct
Part Infectivity - authors are mentioning only the PCR test and quarantine, some words about antigen tests is missing (which are mostly used in some countries)
Minor issues:
line 27 - Coronaviridae italics
line 29- SARS-CoV2 is a virus causing COVID-19 disease
line 41 - authors are citing MERS transmission, however they are explaining the SARS-CoV2 infection. Both viruses are from the same family, however, they share very low similarities.
line 88 - 19 is missing